# Fatigue Behaviors of Joints between Steel Girders with Corrugated Webs and Top RC Slabs under Transverse Bending Moments

**DOI:** 10.3390/ma16062427

**Published:** 2023-03-18

**Authors:** Yun Zhang, Tao Yang, Tingyi Luo, Mingyu Chen, Xiaobin Chen

**Affiliations:** 1Guangxi Beitou Highway Construction and Investment Group Co., Ltd., Nanning 530029, China; 2College of Civil Engineering and Architecture, Guangxi University, Nanning 530004, China; mingx1006@163.com (M.C.); cxb961101@163.com (X.C.); 3Key Laboratory of Disaster Prevention and Structural Safety of Ministry of Education, Nanning 530004, China

**Keywords:** corrugated steel web, transverse bending moment, fatigue performance, fatigue life, fatigue damage

## Abstract

Steel–concrete composite box beams are widely used in bridge engineering, which might bear transverse and longitudinal bending moments simultaneously under vehicle loads. To investigate the fatigue performance of joints between the steel girders and the top reinforced concrete (RC) slabs under transverse bending moments, a reduced scale joint between the weathering steel girder with the corrugated steel web (CSW) and the top RC slab was designed and tested under constant amplitude fatigue loads. Test results show that the joint initially cracked in the weld metal connecting the CSW with the bottom girder flange during the fatigue loading process. The initial crack propagated from the longitudinal fold to the adjacent inclined folds after the specimen was subjected to 7.63 × 10^5^ loading cycles and caused the final fatigue failure. Compared with the calculated fatigue lives in the methods recommended by EC3 and AASHTO, the fatigue performance of the details involved in the joint satisfied the demands of fatigue design. Meanwhile, finite element (FE) models of joints with different parameters were established to determine their effect on the stress ranges at the hot spot regions of the joints. Numerical results show that improving the bending radius or the thickness of the CSW helps to reduce the stress ranges in the hot spot regions, which is beneficial to enhance the fatigue resistance of the investigated fatigue details accordingly.

## 1. Introduction

Steel–concrete composite beams with corrugated steel webs (CSWs) have the advantages of high transmission efficiency of prestress and adequate utilization of mechanical performance of concrete and steel. Hence, composite beams with CSWs have been increasingly used in engineering structures and broadly studied in the past several decades [1,2,3]. Chen et al. [4] experimentally found that the composite box girder with CSWs and trusses showed good ductility. Elkais et al. [5,6] built a new equation to evaluate the ultimate shear strength of plate girders with CSWs based on numerical analysis and found that web thickness was the key factor affecting the shear capacity of the girders. In practice, composite beams with CSWs are prone to fatigue damage due to their complex constructional details under fatigue loads [7]. Li et al. [8] found that the beams with CSWs showed excellent fatigue performance Zcompared to those with flat steel web, and stress concentration is prone to appear in the tension flange at the web-to-flange fillet weld toe [9,10]. The radius of the corrugation curvature and the inclination angle of the inclined plate were the main inducements of the stress concentration at the joint between the CSWs and the top slab [11]. Accordingly, fatigue cracks usually originate in the joint between the CSWs and the top slab [12,13,14]. To predict the potential fatigue failure of composite beams with CSWs, calculation methods for the fatigue life have been investigated by many researchers as well [15,16,17].

The fatigue behaviors of beams with CSWs under longitudinal bending moments have been studied by some researchers so far [18,19]. In practice, composite box beams with CSWs usually bear longitudinal and transverse bending moments simultaneously, and fatigue failure of joints between the steel girders and the top reinforced concrete (RC) slabs is most likely to occur. However, the fatigue performance of the joints under transverse bending moments is still less studied and is worthy of investigation. This paper reports the fatigue tests on a reduced scale joint between the weathering CSW and the top slab subjected to transverse bending moments. Meanwhile, joints with different design parameters were developed to evaluate the effect of design parameters, including the bending radius and the thickness of CSWs on the stress status of the hot spots, aiming to provide bases for the fatigue design of the joints under similar working conditions.

## 2. Test Design

### 2.1. Specimen Design and Materials

A steel–concrete composite box beam bridge is chosen as the prototype structure to design the tested joint specimen, the typical cross-section of which is shown in Figure 1. Perfobond rib (PBL) shear connectors are employed to connect the steel girder with the top RC slab. Due to vehicle loads acting on the top slab, the composite box beams are subjected to longitudinal and transverse bending moments at the same time. Therefore, it is necessary to evaluate the potential risk of fatigue failure of the joint between the steel girder and the RC slab under the transverse bending moment. In the prototype beam, the maximum top slab thickness is greater than 800 mm. Limited by the laboratory conditions, the top slab thickness has to be properly reduced. To ensure that the mechanical status of the joint specimen can reflect that of the prototype beam, three finite element (FE) models of joints with slab thicknesses of 300 mm, 500 mm, and 800 mm were developed in ABAQUS software for sensitivity analysis, and it shows that reducing the slab thickness from 800 mm to 300 mm slightly affects the stress distribution of the joint. Therefore, the top slab thickness of the specimen was determined as 300 mm.

A specimen of the joint between the top RC slab and the steel girder was designed referring to the isolated body plotted in Figure 1, which is enclosed by the dotted line. Twin PBL connectors were used to transfer the shear force between the top slab and the steel girder. Geometries and configuration details of the joint are shown in Figure 2. The RC top slab was 1800 mm long and 1710 mm wide with a thickness of 300 mm. Slab reinforcement cage consisted of two layers of reinforcing bars. The HRB400 hot-rolled ribbed reinforcement with diameters of 20 mm and 16 mm (D20 and D16) was used as the top and bottom longitudinal slab reinforcement, respectively. Meanwhile, the top and bottom transverse slab reinforcements were D20 and D25, respectively, and D12 was used as the tie reinforcement. The bottom transverse slab reinforcement D25 passed through the holes set in the PBL connectors. Q420qDNH weathering steel plates were used to fabricate the PBL shear connectors and the steel girder with CSW, which were 16 mm and 14 mm in thickness, correspondingly. The configuration of the CSW and the arrangement of PBL connectors are depicted in Figure 2c–e. The top and bottom girder flanges were 20 mm and 40 mm in thickness, which were welded with the CSW by the full penetration fillet welds with a leg length of 14 mm. Similarly, the PBL connectors were welded to the top girder flange by fillet welds with a leg length of 16 mm. Mechanical properties of concrete and steel were tested before the formal tests. The uniaxial strength of concrete tested on three 150 mm × 150 mm × 300 mm prisms cured with the specimen under the same conditions was 51.0 MPa. The measured ultimate strength of the steel used for the CSW and the PBL connectors was 611.4 MPa and 633.9 MPa, respectively. Eight predrilled holes of 80 mm in diameter were arranged on both sides of the bottom girder flange, as shown in Figure 2a,b, through which high-strength anchor rods passed to fix the specimen on the test bed.

### 2.2. Test Setup and Loading Scheme

The joint specimen comprises four main fatigue details following EC3 [20], which belong to categories 125, 80, 80, and 71, respectively, and were named details 1–4, as shown in Figure 3. The base metal of the CSW and the weld metal connecting PBL connectors with the top girder flange were classified into details 1 and 3 in sequence, and the welds joining the girder web to the top and bottom girder flanges were classified into details 2 and 4, respectively, in the light of flange thickness. As demonstrated in the completed numerical analysis, the hot spot stress usually appeared near the bend regions from the longitudinal folds to the inclined folds of the CSW, which was also verified by Ibrahim et al. [12]. Therefore, the stress at the bending spots from the longitudinal folds to the inclined folds of the CSW was adopted to determine the fatigue loads in the test. As the hot spot stress on the CSWs of the prototype bridge was approximately 10 MPa under the dead design load, the stress on the CSW of the joint specimen was also taken as 10 MPa following similar conditions. Limited by the laboratory conditions, the loading point on the top slab was 780 mm away from the longitudinal axis of symmetry of the upper girder flange. Then, the minimum fatigue load $P_{\min}$ was deduced as 10 kN. Preliminary analysis shows that detail 2 was prone to fatigue damage compared to the other three details under the transverse bending moments. Therefore, the maximum fatigue load in the fatigue test was determined as $P_{\max}=P_{\min}+\Delta P=110\;\mathrm{kN}$, where ΔP=100 kN is the fatigue load range causing a stress range $\Delta \sigma_{c}=80\;\mathrm{MPa}$ at detail 2, and $\Delta \sigma_{c}$ corresponds to the calculated fatigue life $N_{c}=2\times 10^{6}$ following EC3 code.

Figure 4 shows the test setup for the fatigue tests. The bottom girder flange was fixed on the test floor through eight ground anchor rods, and a spreader beam was arranged on the top surface of the RC slab to ensure the applied load could be uniformly transferred. The MTS servo-actuator with a maximum loading capacity of 500 kN was employed to apply the vertical force on the spreader beam. Figure 5 depicts the loading procedure, which mainly consists of three steps: (i) The preload test will be performed in the multi-stage loading method to eliminate the gap between the specimen and the devices and check the working conditions of the measuring instruments. (ii) Static loading and unloading cycles are to be carried out before the formal fatigue loading. (iii) Fatigue tests with a constant amplitude load range will be conducted at the frequency of 5 Hz. The minimum and maximum fatigue loads were $P_{\min}=10\;\mathrm{kN}$ and $P_{\max}=110\;\mathrm{kN}$. When fatigue loading cycles reach 1 × 10^4^, 5 × 10^4^, 1 × 10^5^, 2 × 10^5^, 5 × 10^5^, 1 × 10^6^, 1.5 × 10^6^, and 2 × 10^6^, static loading and unloading tests are to be performed to evaluate the degradation of the mechanical performance of the joint specimen. (iv) Once no apparent fatigue damage is observed after 2 × 10^6^ loading cycles, the fatigue loading will be continued after appropriately enhancing the maximum fatigue load until the final fatigue failure occurs. The applied force, the vertical deflection of the top slab, and the strains at the measuring points are to be continuously observed during the tests.

## 3. Experimental Results and Discussion

### 3.1. Observations during the Fatigue Loading Procedure

The constant amplitude fatigue loading with a load range ΔP=100 kN was implemented during the fatigue tests. When the loading cycles reached 5 × 10^5^, an initial crack appeared near the weld toe connecting the CSW with the bottom girder flange. When the loading cycles reached 7.63 × 10^5^, the crack propagated from the longitudinal fold to the adjacent inclined folds. By this time, it was hard to keep $P_{\max}$ constant during the loading procedure due to the severe cracking of the weld, and then the fatigue test ended. The failure mode of the joint is shown in Figure 6a. During the whole loading procedure, a primary crack gradually formed on the top slab surface in the longitudinal direction; no fatigue damage was observed for the PBL shear connector and the weld nearby after partially removing the RC top slab, as shown in Figure 6b. Meanwhile, no fatigue damage was observed for the other constructional details.

### 3.2. Strain Development of the CSW

Strain gauges were stuck on the CSW to record the strain development. The measuring points are shown in Figure 2a. Figure 7 shows the development of strain on the CSW under $P_{\max}$ during the first static loading procedure. As localized on the convex side of the CSW, the observation points are all in tension, and the strains tended to increase linearly with the applied load. The strain at CW2 was greater compared to CW1 and CW3, and that of CW5 was also greater than those at CW4 and CW6. It indicates that the hot spot stress did appear in the bending regions from the longitudinal fold to the inclined folds of the CSW.

Figure 8 shows the strain development at the measuring points under $P_{\min}$ and $P_{\max}$ after experiencing specific loading cycles. The strains present the following characteristics: (1) The strains under $P_{\min}$ changed slightly during the cyclic loading process except for the latter loading stage. (2) After an initial several thousand loading cycles, a short fast-increasing stage was observed at most of the measuring points under $P_{\max}$; then, the strains exhibited different developing tendencies. For the measuring points located near the joint between the top girder flange and the CSW, the strains usually kept constant (e.g., CW1 and CW3) or steadily increased (e.g., CW2); however, for the measuring points distributed near the connection between the girder web and the bottom flange, the development of the strains was tightly related to the accumulation of fatigue damage to the welds (e.g., CW4, CW5, and CW6), as shown in Figure 8b. As observed in the test, cracking of the weld toe originated from the bend region of the fold line where CW5 was located. The strain at CW5 decreased with the extension of the crack, while those at CW4 and CW6 increased instead, indicating that stress redistribution occurred. (3) The strains at the measuring points CW2, CW4, CW5, and CW6 all underwent a fast-increasing stage at the end of the fatigue test, which portended the final fatigue failure of the specimen. According to the observed development tendency from Figure 8, the fatigue damage of beam members in practice subjected to fatigue loads could be assessed by monitoring the strain at critical points, for example, the bend points of CSWs.

### 3.3. Strain of PBL Connectors

Strain gauges were stuck on the PBL shear connectors to monitor the strain development, and the measuring points are shown in Figure 2e. The strains at the measuring points of the PBLs under $P_{\max}$ during the first static loading are shown in Figure 9. Clearly, the strains at most of the measuring points show an increasing trend with the applied load. Under the transverse bending moment, the PBL located on the convex side of the CSW was mainly subjected to tensile force, referring to the strains at PBl–PB3, while the PBL on the concave side of the CSW was in compression according to the strains at PB4–PB5. As the stains at PB1–PB3 rank in descending order, it indicates that the PBL near the bend regions of the girder web is subjected to greater uplift force under the transverse bending moments, and fatigue failure is most likely to first occur near PB1.

The strain development at PB1–PB3 under $P_{\min}$ and $P_{\max}$ after designated cyclic loading are shown in Figure 10. It indicates that the strains at PB1 fluctuated during the initial 2 × 10^5^ loading cycles, while those at PB2 and PB3 remained smaller compared to that at PB1. In the following loading procedure, the strains at PB1 dramatically decreased while that at PB2 remained increasing, resulting from the stress redistribution caused by the fracture of the weld. According to the recorded strains, the stress range at PB1 under fatigue loads approached 180 MPa, corresponding to a fatigue life of 6.67 × 10^5^ cycles following the prediction method recommended by EC3. Combined with the observations after the test, no fatigue failure was observed at the weld connecting the PBL with the top girder flange after 7.63 × 10^5^ loading cycles. Therefore, the fatigue performance of detail 3 satisfied the demands of fatigue design.

### 3.4. Degeneration of Flexural Stiffness of the Joint

Static loading tests were conducted on the specimen after completing specific fatigue loading cycles, as described in Section 2.2. Figure 11 shows the variation of the vertical deflection of the top slab at the measuring point D1 (see Figure 2c) under $P_{\max}$. Figure 11a indicates that the secant slopes of the load-deflection curves are nearly identical in the early 5 × 10^5^ loading cycles, although the residual deflection had been produced. Then, the secant slopes of the curves kept decreasing until the fatigue failure occurred. Once the residual deflection was neglected (see Figure 11b), the secant slopes maintained decreasing tendency with the increase of loading cycles.

Figure 12 depicts the development of deflection under $P_{\max}$ without considering the residual deflection. The curve can be approximately divided into three stages: (1) the deflection experienced a short fast-increasing stage during the initial several thousand loading cycles; (2) the deflection almost linearly increased before 5 × 10^5^ loading cycles were attained; (3) when the loading cycles were greater than 5 × 10^5^, the growth rate of the deflection accelerated till the test ended. The measured deflection at D1 under $P_{\max}$ before the fatigue tests and after 7.63 × 10^5^ loading cycles was 3.95 mm and 18.87 mm, respectively, indicating that the flexural stiffness of the specimen degraded significantly. As a result, the deflection of the top slab could be used as an important index to assess the accumulation of fatigue damage.

### 3.5. Assessment of Fatigue Lives

Four fatigue details were included in the specimen following EC3, that is (1) detail 1—the base metal of the CSW; (2) detail 2—the fillet weld joining the PBL with the top girder flange; and (3) details 3 and 4—the fillet welds joining the CSW with the top and bottom girder flanges. The four details can be classified into categories B and C following AASHTO specifications [21]. Table 1 lists the classification of the details and the calculated fatigue life *N*_c_ based on the formulas provided by EC3 and AASHTO, and *N_a_* is the actual loading cycle. The following conclusions can be drawn based on Table 1: (1) Although the fatigue failure of the specimen is regarded to happen after 7.63 × 10^5^ loading cycles, the actual fatigue lives of details 2–4 are all greater than the calculated fatigue lives following EC3 and AASHTO. Meanwhile, it is concluded that fatigue failure will not occur for detail 1 as the actual stress range was less than the threshold values. As a result, the four details all satisfy the demands of fatigue design. (2) The weld joining the PBL shear connectors to the top girder flange could satisfy the demands of fatigue design under the applied load. Potential fatigue failure of PBL is most likely to occur at the location right over the bend region of the fold line of the CSW. (3) The calculated fatigue lives of the details based on EC3 are usually less than that on AASHTO, indicating that the calculated fatigue life following the EC3 code is relatively conservative.

## 4. Finite Element Modeling Method

### 4.1. Modeling Methods

As greater stress ranges will cause premature fatigue failure of fatigue details, reducing the stress ranges of vulnerable fatigue details is a practical approach. For the joint between the steel girder and the top RC slab under transverse bending moments, parameters such as the bend radius of fold lines and web thickness of the CSW will directly affect the stress distribution. The influence of these parameters on the stress status of joints is to be evaluated in the finite element (FE) method to provide a reference for optimizing fatigue design.

A finite element (FE) model was built in the ABAQUS software referring to the tested specimen. Half of the model was built considering the symmetry of geometry and boundary conditions to enhance the computational efficiency. The concrete damaged plasticity model (CDPM) was used to simulate the mechanical behavior of concrete, referring to Yang et al. [22,23]. The bilinear model was adopted to simulate the constitutive relationships of steel and reinforcement. The connection between steel elements was simulated using the “tie” command, and the rebars interacted with concrete in a built-in way. The surface-to-surface contact between the PBLs and the concrete was used, and the friction coefficient was set as 0.25. The mesh size of the concrete slab was about 40 mm, and those of the PBL and the girder flanges were 20 mm. Additionally, the mesh size in partial regions of the CSW near the top and bottom girder flanges was refined as 5 mm to improve the calculation accuracy. The meshed FE model is shown in Figure 13a. Predrilled holes were built in the bottom girder flange, and the regions around the holes were constrained entirely, as shown in Figure 13b. A bearing plate was set on the top slab surface, on which a concentrated load is applied in the displacement control method.

### 4.2. Validation of Numerical Results

The feasibility of the proposed modeling method was validated by comparing it to the test results. The stress nephogram of the FE model is shown in Figure 14, showing that the stress distribution law of the CSW presented the same characteristics as those observed in the test. Figure 15 shows the comparison of the deflection of the top slab at D1 and the stress status at the critical points obtained from the test and FE analysis (FEA). It indicates that the curves meet well with each other. Therefore, the FE modeling method is reasonable for assessing the mechanical behaviors of the joint under the transverse bending moments.

## 5. Parametric Analysis

Seven FE models were developed in the proposed method to evaluate the effect of different design parameters, i.e., the bending radius of the CSW, the web thickness, and the web corrugation, on the stress status of fatigue details. The design parameters of each model are tabulated in Table 2. Figure 16 depicts the schematic diagram of the design parameters.

The stress amplitude of four fatigue details was calculated, referring to the EC3 and the AASHTO specifications. Table 3 lists the calculated results and the constant amplitude fatigue threshold value $\Delta \sigma_{L}$ stipulated by EC3 and AASHTO. When the stress ranges are less than $\Delta \sigma_{L}$, the fatigue life can be infinite accordingly. The comparison shows that the fatigue stress ranges of details 1 and 3 for each specimen are less than $\Delta \sigma_{L}$ in most cases, which means fatigue failure will not happen here with a high probability. Therefore, the following discussion will mainly focus on the stress status of details 2 and 4.

### 5.1. Discussion of Stress Distribution for PBL

During the comparisons of the stress status on PBLs, it is found that the von Mises stress distribution is mainly concentrated on the PBL away from the loading point, as shown in Figure 17. Therefore, the discussion is developed around this PBL shear connector. The stress distribution of the PBL is likewise affected by the bend radii and corrugations of the web, except for web thickness, according to the FE analysis results. Figure 18 compares the stress distribution for PBL shear connectors with different radii of CSWs. It can be seen that the stress distribution is mainly concentrated above the longitudinal fold of CSWs and is negligible above the inclined fold. The maximum stress of S2 in the PBL is 100 MPa, which is 4.2% and 6.4% greater than S1 and S3, respectively. Therefore, the decrease of bend radii of CSWs is adverse to the stress status of the PBL shear connectors in the joints. Figure 19 compares the vertical von Mises stress nephogram of the PBL with different web corrugations. Similarly, the stress distribution is mainly concentrated above the longitudinal fold. The maximum stress of S6 is 29.2% and 11.4% greater than S1 and S7, respectively. As a result, increasing the web corrugation is beneficial for the stress status of the PBL connectors.

### 5.2. Optimization of Fatigue Design

#### 5.2.1. Effects of the Bend Radii of CSWs

S1–S3 were designed to assess the influence of bend radii of CSWs on the stress status of the most adverse positions on the CSW, i.e., CW2 and CW5. As shown in Figure 20, the stress at CW2 and CW5 presents a linearly developing tendency. When the load reaches 110 kN, the stress at CW2 of S2, S1, and S3 are 137 MPa, 127 MPa, and 120 MPa, respectively, and the stress at CW5 are 216 MPa, 197 MPa, and 182 MPa, in sequence. Therefore, increasing the bend radius of the CSW is beneficial to reducing the stress range and further improving the fatigue performance of the constructional details. Figure 21 shows the fatigue lives of details 2 and 4 of S1 to S3, which were calculated in accordance with EC3 and AASHTO. It can be seen that the fatigue life calculated by the EC3 code is more conservative. When the bend radius is increased from 100 mm to 200 mm, the fatigue lives of details 2 and 4 increased by 130.2% and 167.1%, respectively. Therefore, the fatigue lives of details 2 and 4 will increase with the increasing bend radii.

#### 5.2.2. Effects of the Web Thickness

The load-stress curves at CW2 and CW5 of S1, S4, and S5 are shown in Figure 22. The maximum stresses at CW2 of S4, S1, and S5 under the load of 110 kN are 150 MPa, 127 MPa, and 115 MPa, respectively, and those at CW5 are 235 MPa, 197 MPa, and 167 MPa, in turn. The results show that the web thickness significantly affects the stress near the hot spot regions of the CSWs. Properly improving the girder web thickness is advantageous for the fatigue performance of the joints between the steel girder and the RC slab. Figure 23 compares the fatigue lives of details 2 and 4 for models with various web thicknesses. It can be seen that the fatigue lives of details 2 and 4 dramatically increase with the increasing web thickness. When the web thickness is increased from 10 mm to 18 mm, the fatigue lives of details 2 and 4 increased by 254.9% and 279.5%, respectively, according to EC3 and AASHTO. Therefore, increasing the web thickness is also beneficial for the fatigue life of detail 2 and 4.

#### 5.2.3. Effects of the Corrugations of the Web

Figure 24 shows the strain development at CW2 and CW5 of S1, S6, and S7. It shows that the stress at CW2 of S6, S7, and S1 is 151 MPa, 134 MPa, and 127 MPa under the load of 110 kN, respectively, and those at CW5 are 216 MPa, 204 MPa, and 197 MPa, in turn. The stress at CW2 of S1 is 3.4% and 8.8% smaller than that of S7 and S6, respectively. Therefore, enhancing the web corrugation will reduce the stress at CW2 and CW5. Note that when the web corrugation is greater than 1600 mm, e.g., S1 and S7, the decreasing tendency in the stress is weakened, and the influence of the web corrugations on the stress status becomes limited. Figure 25 compares the calculated fatigue lives of details 2 and 4. It indicates that those of S6 decreased by 31.3% and 24.6%, respectively, compared to S1, as well as 30.2% and 15.8% compared to S7, respectively. Therefore, the decrease in web corrugation will lead to a reduction in the fatigue life of the fatigue details.

## 6. Conclusions

Fatigue tests were conducted on a joint between the steel girder with CSW and the top RC slab under transverse bending moments; then, FE models of joints with different design parameters were built and analyzed to determine their effect on the fatigue performance of hot spot regions. The following conclusions have been obtained:(1)The weld joining the bottom girder flange and the CSW initially cracked at the bend regions between the longitudinal and inclined folds of the CSW after 5 × 10^5^ loading cycles during the fatigue test. The crack gradually propagated from the longitudinal fold to the two adjacent inclined folds. Stress on the steel girder was redistributed with the accumulation of fatigue damage, and the vertical deflection of the top slab gradually increased due to the degradation of the flexural stiffness, which could be used as an indicator to assess the accumulation of fatigue damage.(2)The constructional details 1–3 comprised in the tested joint all suffered 7.63 × 10^5^ cyclic loading and no fatigue damage was observed, while initial fatigue failure was found for detail 4 after 5 × 10^5^ loading cycles. It demonstrates that the actual fatigue lives of the four details are longer than the calculated lives in the prediction methods provided by EC3 and AASHTO. As only one joint specimen was tested, the fatigue performance of details involved in the joints needs to be validated by more experimental results.(3)FE models were built referring to the tested specimen. The PBL located in the tensile regions mainly suffered uplift forces according to the numerical analysis, and stress concentration is most pronounced on the PBL just above the longitudinal fold of the web. The maximum stresses of the CSWs are concentratedly distributed within the longitudinal fold near the top and bottom girder flanges. As fatigue failure is most likely to first occur in the hot spot regions, more attention should be paid to the stress status in these regions in fatigue design.(4)Numerical analysis results show that increasing the web thickness is beneficial to reducing the stress ranges in the hot spot regions. Meanwhile, the stress range in the hot spot regions and the maximum stress of the PBL decrease with the increasing bend radii of CSWs. Greater web corrugations will improve the fatigue resistance and reduce the maximum stress of the PBL, while this advantageous effect becomes limited when the web corrugation is greater than 1600 mm.

## Figures and Tables

**Figure 1 materials-16-02427-f001:**
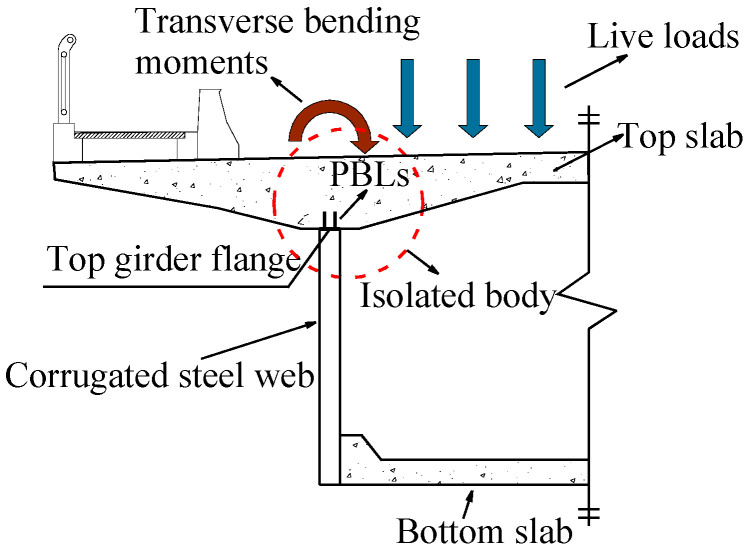
Typical cross-section of the prototype composite beam.

**Figure 2 materials-16-02427-f002:**
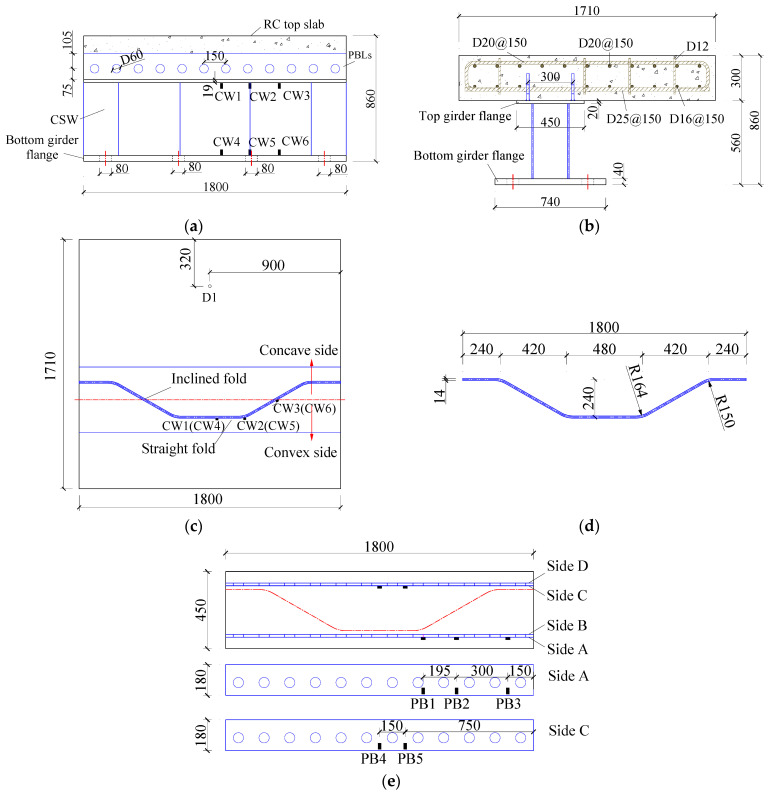
Design details and layout of the measuring points: (**a**) front view; (**b**) side view; (**c**) top view; (**d**) dimensions of the CSW; (**e**) measuring points on the PBL connectors.

**Figure 3 materials-16-02427-f003:**
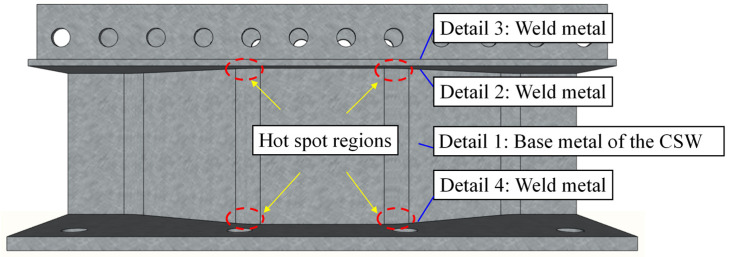
Schematic diagram of the fatigue details in the joint.

**Figure 4 materials-16-02427-f004:**
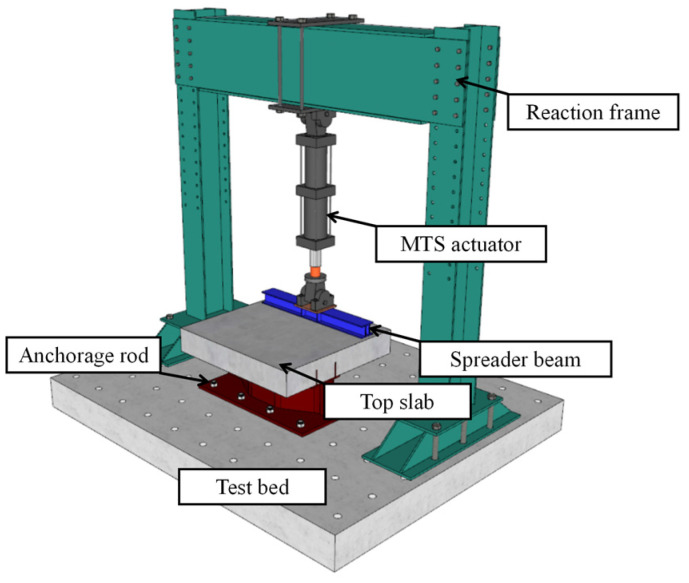
Test setup.

**Figure 5 materials-16-02427-f005:**
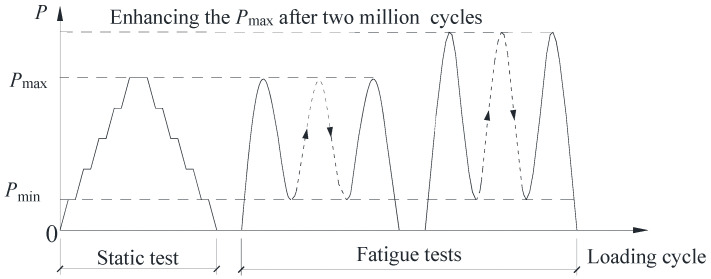
Loading scheme.

**Figure 6 materials-16-02427-f006:**
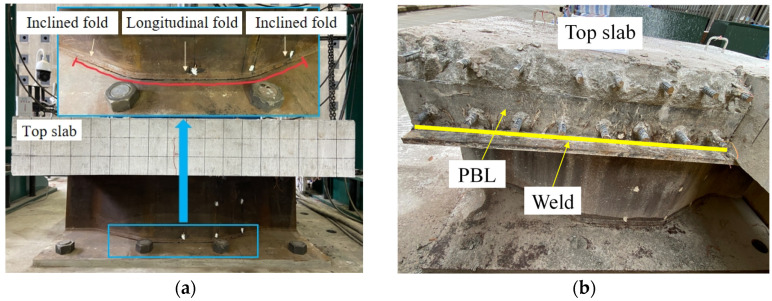
Failure modes: (**a**) cracking of the weld; (**b**) partial demolition of the top slab.

**Figure 7 materials-16-02427-f007:**
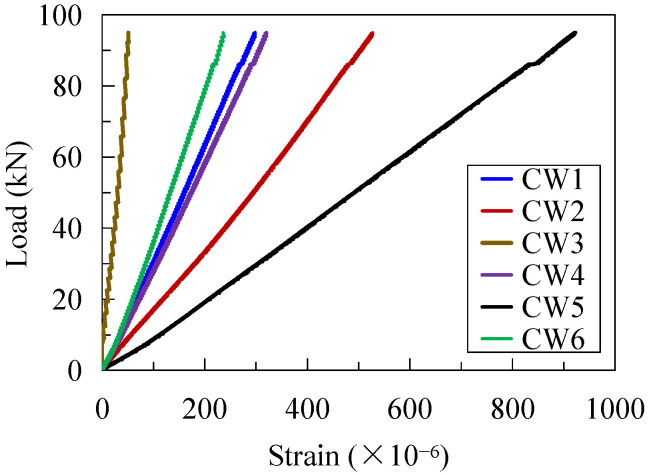
Strain development of the CSW under $P_{\max}$.

**Figure 8 materials-16-02427-f008:**
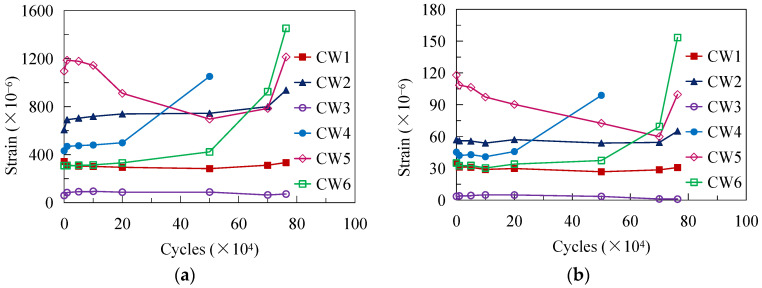
Strain development under cyclic loading: (**a**) *P*_min_; (**b**) *P*_max_.

**Figure 9 materials-16-02427-f009:**
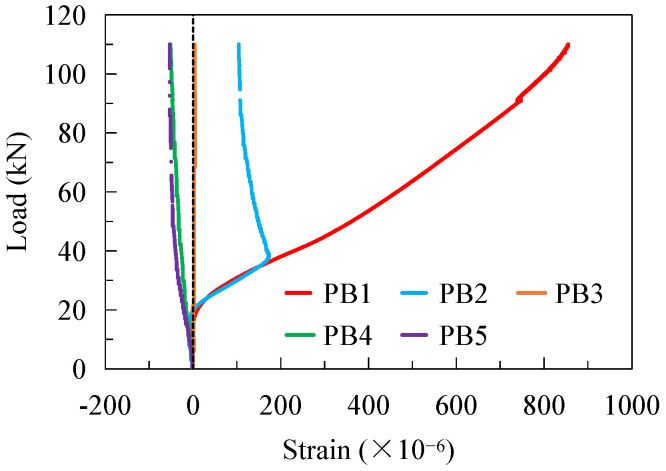
Strains on the PBL connectors.

**Figure 10 materials-16-02427-f010:**
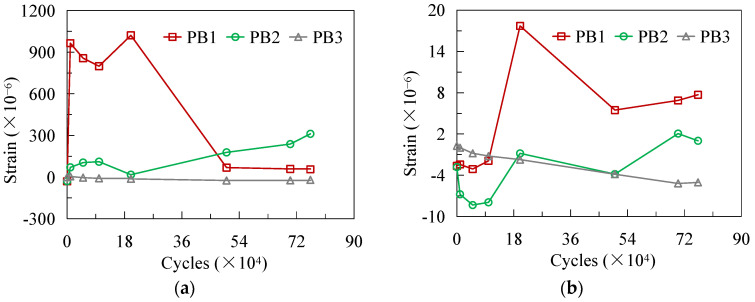
Strain development of PBL connectors under cyclic loading: (**a**) *P*_min_; (**b**) *P*_max_.

**Figure 11 materials-16-02427-f011:**
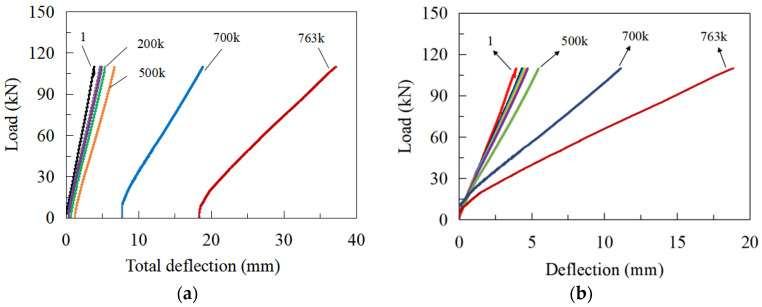
Load-deflection curves under *P*_max_: (**a**) with residual deflection; (**b**) without residual deflection.

**Figure 12 materials-16-02427-f012:**
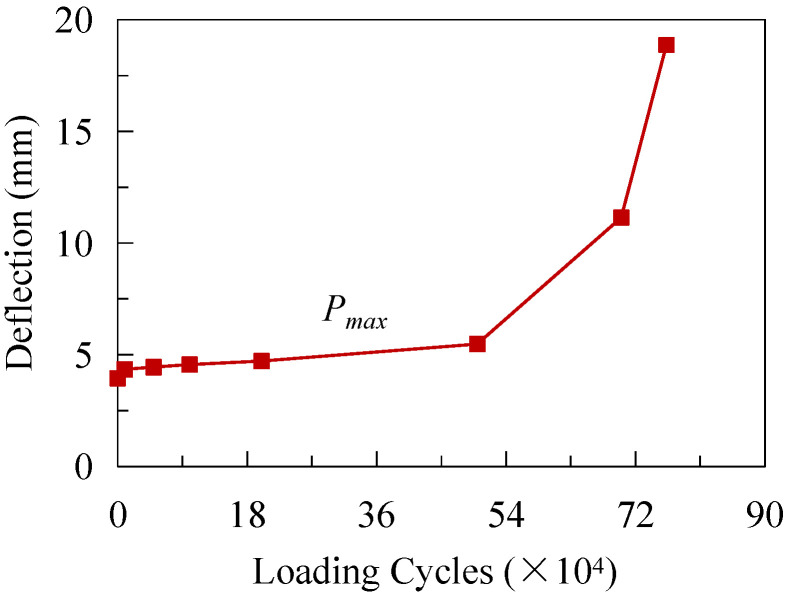
Deflection development under fatigue loading.

**Figure 13 materials-16-02427-f013:**
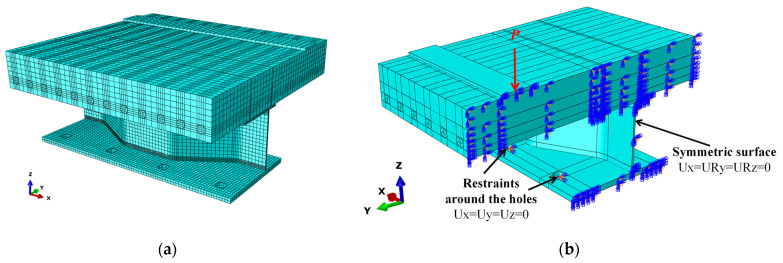
FE model: (**a**) mesh division; (**b**) boundary condition.

**Figure 14 materials-16-02427-f014:**
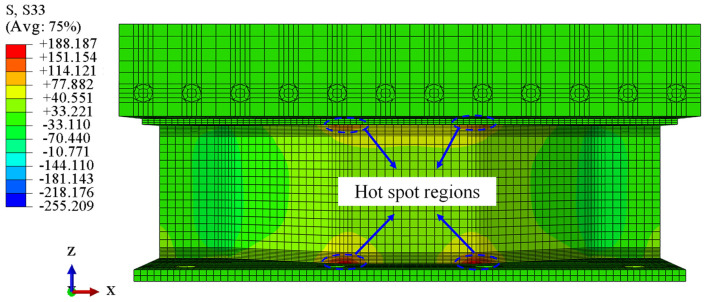
Stress nephogram of the FE model. (Unit: MPa).

**Figure 15 materials-16-02427-f015:**
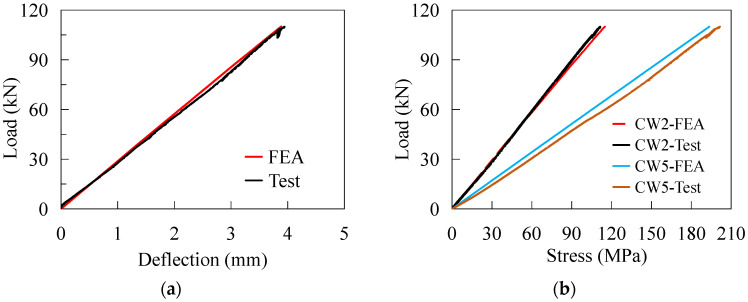
Comparisons of numerical and experimental results: (**a**) load-deflection curves; (**b**) stress at CW2 and CW5.

**Figure 16 materials-16-02427-f016:**
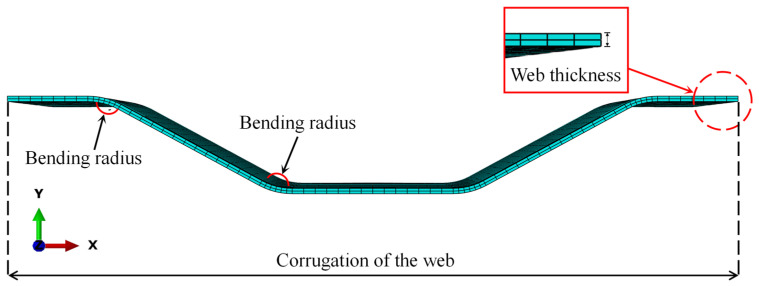
Schematic diagram of the design parameters.

**Figure 17 materials-16-02427-f017:**
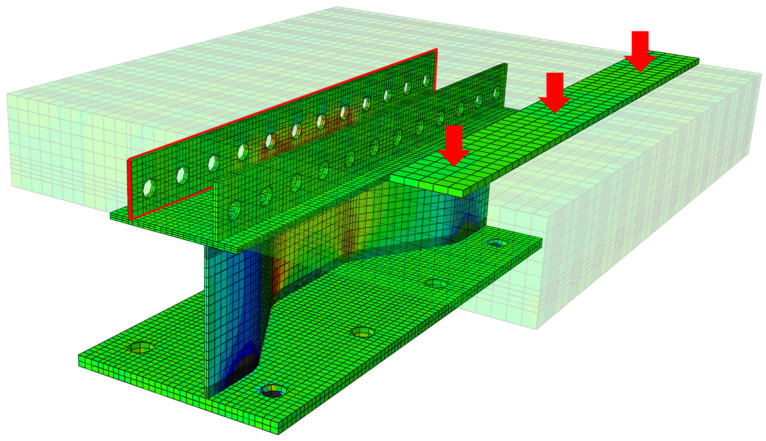
Stress status of the PBL under transverse bending moments.

**Figure 18 materials-16-02427-f018:**
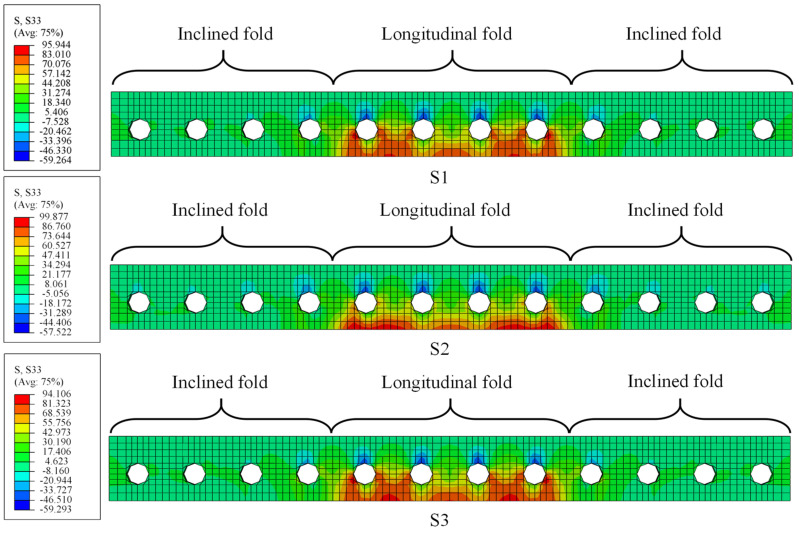
Stress distribution of PBL shear connectors for models with different radii of CSWs.

**Figure 19 materials-16-02427-f019:**
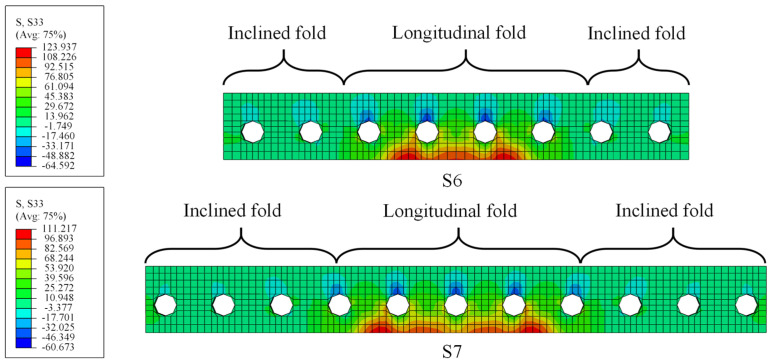
Stress distribution of PBL shear connectors for models with different web corrugations.

**Figure 20 materials-16-02427-f020:**
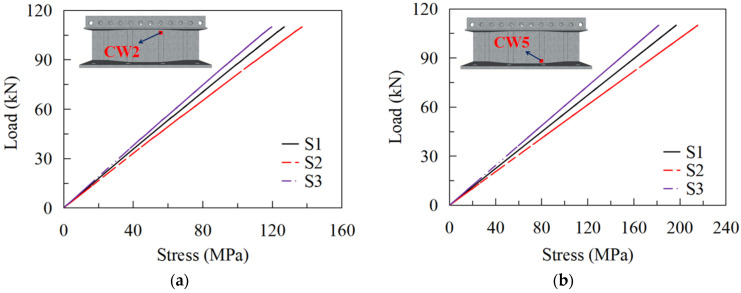
Load-stress curves for models with different bend radii: (**a**) CW2; (**b**) CW5.

**Figure 21 materials-16-02427-f021:**
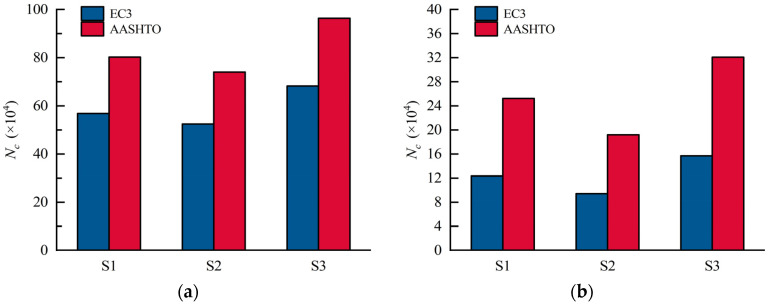
Fatigue lives of S1, S2, and S3 in different codes: (**a**) detail 2; (**b**) detail 4.

**Figure 22 materials-16-02427-f022:**
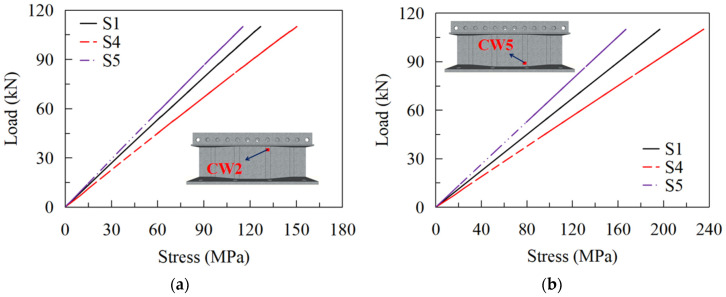
Load-stress curves for models with different web thicknesses: (**a**) CW2; (**b**) CW5.

**Figure 23 materials-16-02427-f023:**
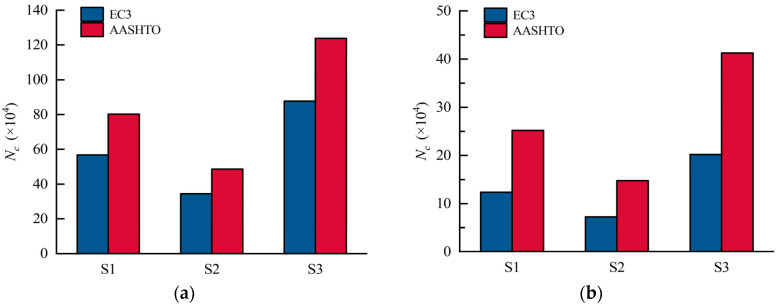
Fatigue lives of S1, S4, and S5 in different codes: (**a**) detail 2; (**b**) detail 4.

**Figure 24 materials-16-02427-f024:**
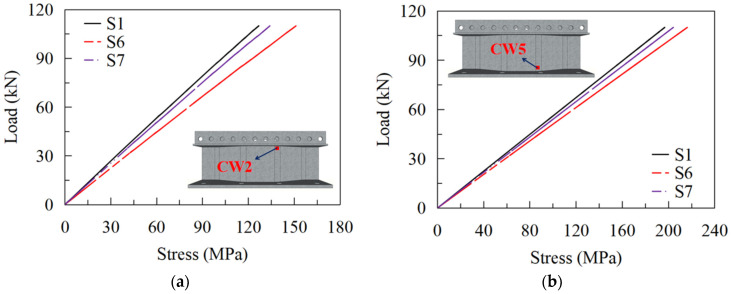
Load-stress curves for models with different corrugations of the web: (**a**) CW2; (**b**) CW5.

**Figure 25 materials-16-02427-f025:**
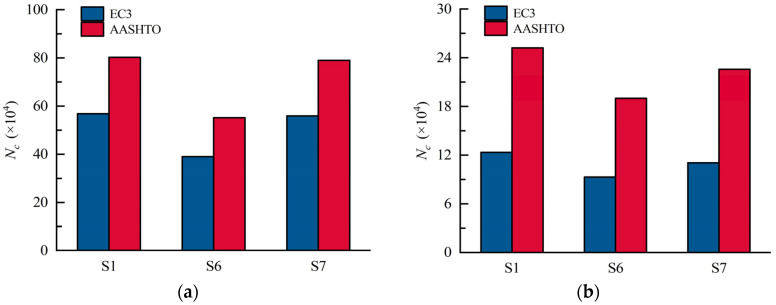
Fatigue lives of S1, S6, and S7 in different codes: (**a**) detail 2; (**b**) detail 4.

**Table 1 materials-16-02427-t001:** Fatigue life assessment of the fatigue details.

Design Codes	Detail ID	Detail Category	Δ*σ*_1_ (MPa)	*N_c_* (×10^4^)	*N_a_* (×10^4^)
EC3	1	125	27	∞	76.3
	2	80	127	49.9	76.3
	3	80	154	28.0	76.3
	4	71	197	9.3	50.0 *
AASHTO	1	B	27	∞	76.3
	2	C	127	70.6	76.3
	3	C	154	39.6	76.3
	4	C	197	18.9	50.0 *

Note: “∞” means the infinite fatigue life in theory as the actual stress range is less than the threshold values specified by EC3 and AASHTO; the data with an asterisk represents the loading cycles when the initial crack of weld metal was observed.

**Table 2 materials-16-02427-t002:** Parametric details of the FE models.

ID	Bending Radius of the CSW (mm)	Web Thickness (mm)	Corrugations of the Web (mm)
S1	150	14	1800
S2	100	14	1800
S3	200	14	1800
S4	150	10	1800
S5	150	18	1800
S6	150	14	1200
S7	150	14	1600

**Table 3 materials-16-02427-t003:** Stress amplitude of four details of FE models.

Codes	Detail ID	Detail Category	Δ*σ*_L_ (MPa)	Δ*σ* (MPa)						
				S1	S2	S3	S4	S5	S6	S7
EC3	1	125	51	30	31	32	45	23	49	38
	2	80	32	122	125	114	144	105	138	122
	3	80	32	59	66	58	62	55	93	77
	4	71	29	179	196	165	214	152	197	186
AASHTO	1	B	110	30	31	32	45	23	49	38
	2	C	69	122	125	114	144	105	138	122
	3	C	69	59	66	58	62	55	93	77
	4	C	69	179	196	165	214	152	197	186

## Data Availability

Not applicable.

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
