# Peer review of "Fatigue Behaviors of Joints between Steel Girders with Corrugated Webs and Top RC Slabs under Transverse Bending Moments"

_materials, 2023, doi:10.3390/ma16062427_

Round 1
Reviewer 1 Report
|
No. |
Comment |
|
1 |
As fatigue tests give scattered results and conclusion on fatigue cannot be based on the results of one sample. This manuscript tested only one reduced scale joint of corrugated steel web and RC top slab under fatigue loading. Therefore, the verification of the model and the conclusions of this manuscript should be ensured by testing additional samples. |
|
2 |
Make sure you describe briefly the tests done by the references you cited in your text (i.e. summarize their conclusions in your manuscript not just cite without elaborating) For example, in the introduction section, references (1-4) are not adequately described |
|
3 |
‘’Preliminary analysis shows that detail 2 was prone to fatigue damage compared to the other three details under the transverse bending moment’’ This preliminary analysis needs to be shown in the manuscript in a more detailed way to be able to understand the reason behind using this stress range |
|
4 |
“the spreader beam arranged on the top of the RC slab lay in the location where was 780 mm from the central axis of the corrugated steel web.” Why was this distance selected instead of exactly the positioning the spreader beam at the centre? |
|
5 |
Why was there a static loading before the fatigue stage? This needs to be addressed. Could this static loading cause an additional degradation in the sample in addition to the fatigue loading? How can one distinguish the degradation in flexural strength caused by the fatigue alone (without the effect of static loading). Please elaborate in the manuscript |

Reviewer 2 Report
Review report: Fatigue behaviors of connection between corrugated steel web and top slab under transverse bending moments. Work is presented well with good publishing quality and can be accepted after the following minor corrections:
1. The abstract section is presented well but add quantitative values at the end of the abstract section.
2. Novelty of the work should be discussed in separate section.
3. Introduction section is written very poorly. It is difficult to get any research gap from introduction section. Rewrite and add the application of the work and add key published work and try to make a bridge between current and previous published work.
4. Add the experimental setup.
5. Provide the clear information about fatigue loading.
6. The surface shown in Fig. 5 need detailed analysis.
7. Discuss the mechanism of the failure which is the main part of the manuscript.
8. The model prepared for study need clear information in terms of nots elements and also about the boundary conditions.
9. Parametric study is presented just like a technical report. Add the technical discussion part separately.
10. Shorten the length of the conclusion section.
Reviewer 3 Report
· The informal language is not suitable and should be improved extensively. The article needs major grammatical and syntax improvements. Use of English service center is recommended. Several sentences are not clear and understandable.
· Majority of the qualitative statements should be modified for quantified result comparisons.
· The introduction needs to be revised for higher quality language. The author mentioned some works without stating about the contributions, pros and cons and the how the current work would address.
· The purpose of the article should be clarified in details, why and where this study could be beneficent, more in depth conclusion should be provided.
· The authors mentioned “therefore, many researchers paid more attention to this issue. It was found that the radius of the corrugation curvature and the inclination angle of the inclined plate were the main inducements of the stress concentration at the corner of the corrugated steel webs.” The following references should be added for comprehensiveness of this statement 1) Behavior prediction of corrugated steel plate shear walls with openings. 2) Analysis and design recommendations for corrugated steel plate shear walls with a reduced beam section 3) Seismic behavior investigation of the corrugated steel shear walls considering variations of corrugation geometrical characteristics. 4) Parametric Computational Study on Butterfly-Shaped Hysteretic Dampers. 5) Investigation of seismic behavior of corrugated steel shear walls considering variations of corrugation geometrical characteristics
· The selection of the specimen, geometrical properties and malarial properties should be justified
· The loading protocol should be clarified and codes should be determined.
· Any Equations and figures taken from other works should be reestablished and referenced
· The modes of the behavior should be elaborated
· For the verification study, the mesh sensitivity and element selection should be elaborated.
Round 2
Reviewer 1 Report
|
No. |
Comment |
|
1 |
The authors did not address the fact that their conclusions on fatigue behavior such as the cracking pattern, fatigue damage, and fatigue life (points 1 and 2 in the Conclusions section) are based on the experimental fatigue results. These fatigue conclusions are drawn based on the results of one sample only. This makes the results unreliable. Therefore, more samples needs to be tested to be able to draw these conclusions. Otherwise, I suggest shifting the focus of the manuscript to the results of the FE model. |
|
2 |
The validation of the FE model needs to be addressed in a separate section with samples loaded under static load up to failure. |
|
3 |
The FE parametric analysis is poorly presented. Consider presenting each parameter (Bending radius of the CSW, Girder web thickness, and Girder web height) as a sub-section. I also suggest expanding the parametric study by studying the effect of more parameters. |
Reviewer 3 Report
The informal language is not suitable and should be improved extensively. The article needs major grammatical and syntax improvements. Use of English service center is recommended. Several sentences are not clear and understandable.
· · The introduction needs to be revised for higher quality language. The author mentioned some works without stating about the contributions, pros and cons and the how the current work would address..
· The authors need to improve the article by most recent work. It is mentioned “therefore, many researchers paid more attention to this issue. It was found that the radius of the corrugation curvature and the inclination angle of the inclined plate were the main inducements of the stress concentration at the corner of the corrugated steel webs.” The following references should be added for comprehensiveness of this statement 1) Behavior prediction of corrugated steel plate shear walls with openings. 2) Analysis and design recommendations for corrugated steel plate shear walls with a reduced beam section 3) Seismic behavior investigation of the corrugated steel shear walls considering variations of corrugation geometrical characteristics. 4) Parametric Computational Study on Butterfly-Shaped Hysteretic Dampers. 5) Investigation of seismic behavior of corrugated steel shear walls considering variations of corrugation geometrical characteristics
